# A Fully-Encircled Polymerized Microfiber Bragg Grating by 3D Femtosecond Laser Nanofabrication

**DOI:** 10.3390/ma15217753

**Published:** 2022-11-03

**Authors:** Fei Xie, Lili Liang, Kang Yang, Sumei Jia, Zhihui Wang, Li Li, Wei Wang, Miaomiao Wang, Guoyu Li, Yan Li

**Affiliations:** Hebei Key Laboratory of Optical Fiber Biosensing and Communication Devices, Institute of Information Technology, Handan University, Handan 056005, China

**Keywords:** two-photon polymerization, fully encircled, microfiber Bragg grating, units pitch, sensitivity

## Abstract

Through the merits of the arbitrary three-dimensional (3D) fabrication ability and nanoscale resolution of two-photon polymerization, we demonstrated a fully encircled polymerized microfiber Bragg grating using 3D femtosecond laser nanofabrication. In order to generate strong enough polymer Bragg grating units around the microfiber surface, and to possess a possible smaller unit pitch and structure size, the composition of photoresist and grating dimensions were both experimentally optimized. A fast-curing, high-adhesion, great-heat-resistant acrylate monomer EQ4PETA was chosen as the cross-linking element, and a high-efficiency photoinitiator DETC was used. Along the tapered microfiber with a diameter of 2 microns, dozens of grating units of 300 nm thickness were successively fabricated. The resonance wavelength was approximately 1420 nm, with a unit pitch of 1 μm, slightly different with varying unit pitches. The refractive index sensitivity reached up to ~440 nm/RIU, which is much higher than other microfiber grating sensors. We also measured the temperature and strain sensitivity of this fully encircled microfiber Bragg grating, and this was estimated at 88 pm/°C and 6.3 pm/µε. It is foreseeable that with the continuous progress of fabrication technology, more highly integrated functional optical devices will emerge in the future.

## 1. Introduction

Featured with an extraordinary compactness down to several micrometers, large evanescent fields, great flexibility, and compatibility with conventional fiber [1,2,3,4], microfiber gratings have attracted extensive attentions in optical communication, sensing detection, nonlinear optics, and biomedical fields [5,6]. Microfiber grating devices with diameters of several micrometers have recently attracted enormous research interest, due to the compactness and flexibility, large evanescent fields, manageable large dispersion, and compatibility with conventional fibers [7]. Especially, the large evanescent field strengthens the interaction between the light and the environment, which induces a more sensitive performance of the microfiber Bragg grating than the conventional fiber Bragg gratings. However, it is not easy to achieve the same refractive index (RI) modulation as conventional fiber Bragg gratings on the surface of microfiber with a diameter of only several microns. As the fiber diameter decreases, the size of the doped fiber core also decreases. The optical signal is coupled out of the fiber core to the air around the microfiber, and forms a strong evanescent field. The thinner the fiber diameter, the stronger the evanescent field intensity. The traditional fiber Bragg gratings inscription method with deep ultraviolet (DUV) 193 nm excimer laser could only induce periodic RI modulation in the doped fiber core, but was unable to do anything about the evanescent field in the free space, which is the key reason why it is difficult to form a high-efficiency cumulative modulation effect with the tapered microfiber. Moreover, methods of femtosecond laser inscription [8,9], ultraviolet (UV) laser irradiation [10], and focused ion beam (FIB) milling [11] are not only less efficient, but also cause physical damage to the microfiber, which leads to the microfiber Bragg gratings being easily broken down [12]. Consequently, in order to form an effective fiber grating modulation effect on the microfiber, the modulation of evanescent field should not be ignored.

A new option is to grow a series of periodic modulation units outside and around the microfiber, to modulate the evanescent field periodically with additive manufacturing methods. Femtosecond direct laser writing (Fs-DLW), or two-photon polymerization (TPP) [13,14], as an outstanding representative, has been demonstrated to be an optimal choice for fabricating 3D high-quality optical devices at the micro/nano scale [15]. Benefitting from the two-photon/multi-photon nonlinear absorption of the photoinitiator in the photoresist, the monomer or oligomer molecules undergo radical or cationic polymerization reactions in the confined femtosecond laser focus region, which is a phase transition from liquid to solid [16]. The limit spatial feature size achieved reaches up to the order of ~100 nm at present. The excellent 3D manufacturing capability ensures that the periodic modulation structures can be created freely around the fiber, perfectly matching the practical requirements of the fiber grating devices. Therefore, the polymer microfiber Bragg grating (μ-PFBG), based on femtosecond laser nanofabrication, has been developed since then, and may become a promising and useful method for μ-PFBG fabrication in the future. The Yiping Wang group from Shenzhen University proposed a μ-PFBG device using the TPP method in 2018 [11]; a series of cuboids with a thickness of 760 nm was fabricated on the two sides of the microfiber with a diameter of 1.7 μm, grating unit pitch of 1.07 μm, and RI sensitivity of approximately ~207 nm/RIU. In the same year, this group demonstrated another suspended polymer fiber Bragg grating (PFBG) in a single-groove silica tube spliced between two single-mode fibers for temperature sensitivity; a sensitivity of −220 pm/°C was achieved over a temperature range of 24 °C to 40 °C [17]. Later, the group proposed an improved helical microfiber Bragg grating for RI sensing, with a special helical structure around a microfiber with a diameter of 2.44 μm [18]. The grating pitch was approximately 1.1 μm, with a thickness of 516 nm, and the RI sensitivity was enhanced to ~229 nm/RIU. At present, it is not easy to generate tens or hundreds of polymerized grating units and keep them stable without collapsing on such a thin microfiber surface during the complex TPP nanofabrication steps. The thinner the grating unit, the smaller the contact area with the microfiber, leading to even worse stability. A compromise must be made between the limit grating-unit size and structure stability. In addition, more effort should be made in photoresist components to improve the molecular crosslinking density, to enhance the structural stability. 

In this work, a novel fully encircled μ-PFBG was proposed, featured with the grating units fully wrapping around the microfiber. The thin grating units have a record thickness of 300 nm, and the design of fully wrapping of the microfiber increases the contact area with the microfiber as much as possible, improving the stability of the structure. In addition, a fast-curing, high adhesion, great heat-resistant acrylate monomer (4) Ethylene oxide pentaerythritol tetraacrylate (EQ4PETA) and a photoinitiator 7-diethylamino-3-thenoylcoumarin (DETC) with high initiation efficiency, were chosen as the photoresist components. We characterized the transmission and reflection spectrum of this novel fully encircled μ-PFBG. The RI sensitivity of the fully encircled μ-PFBG was enhanced up to ~440 nm/RIU, which is much higher than other microfiber grating sensors. Furthermore, the temperature and strain sensitivity obtained were 88 pm/°C and 6.3 pm/µε, respectively. Results show that the proposed scheme is helpful for improving the strength and stability of the structural μ-PFBG device.

## 2. Materials and Methods

The schematic diagram of the experimental setup of femtosecond laser nanofabrication system is illustrated in Figure 1a. A femtosecond laser beam (532 nm, 210fs, 80 MHz) is controlled by a mechanical optical shutter and then focused on the prepared tapered microfiber surface with an oil-immersion 100× oil immersion objective lens (UplanSApo, Olympus, Tokyo, Japan) with a numerical aperture (NA) of 1.4. The microfiber is tapered from a single-mode fiber with a homemade flame-heating tapering system, which is mainly composed of a hydrogen-oxygen flame nozzle fixed onto a piezo stage, and two fiber holders separately fixed onto two micro-displacement stages in opposite directions. The relative motion between the flame and the tapered fiber is achieved with a self-written computer program [19]. The diameter of the microfiber is less than 2 μm, to guarantee sensitivity to the environment. The prepared microfiber is immersed in a drop of photoresist, and sandwiched between the cover and slide glasses, which are mounted on a three-axis piezo-electric stage (P563.3CD, Physik Instrumente, Karlsruhe, Germany). A charge-coupled device (CCD) camera (MER-132-43U3M-L, Daheng Optics, Shanghai, China) is used to observe the state of the focus spot and to monitor the fabrication process. An optical broadband source (BBS) and an optical spectrum analyzer (OSA) are used to observe the reflection spectrum changes during the polymer Bragg gratings fabrication. 

As was illustrated in Figure 1b, dozens of neatly arranged polymer thin cuboids were sequentially fabricated and fully wrapped around the microfiber, generating a fully periodic modulation of the optical signal in the evanescent field. For example, on a microfiber with a diameter of 2 μm, the lateral dimensions of the grating units had to be slightly larger than the microfiber diameter, which was defined as 3 μm × 3 μm, to extend to the evanescent field, forming a modulation waveguide. To obtain a possible smaller grating size, different thicknesses of grating units were experimented with. Figure 1c,d shows two scanning electron microscope (SEM) images of a fabricated fully encircled μ-PFBG at different shooting angles, which has an ultra-thin unit thickness of 270 nm and a unit pitch of 1μm. The fabrication femtosecond laser power of 532 nm was fixed to 1.5 mW, with 5% intrinsic fluctuation, and the scanning speed was approximately 60 μm/s. However, the polymer units were too thin and not strong enough, so an optimal thickness of 300 nm was experimented with, and was defined as a safer choice. Even so, the ultra-small size achieved in this work is more advantageous than that of previously reported work, which has a polymer slice unit thickness of 760 nm [11], 516 nm [18]. On the premise of ensuring optical device performance, the reduction of size is conducive to the further development of miniaturization and integration. In addition, to further improve the stability of the thin grating units, we carefully selected and tested a new photoresist composition with a fast-curing, high adhesion, good heat-resistance acrylate monomer EQ4PETA (Sartomar, Guangzhou, China) and a high-efficiency two-photon absorption (TPA) photoinitiator DETC (Acros Organics, Geel, Belgium) of 0.5% mass fraction. The RI of the polymerized photoresist was approximately 1.53 at 1550 nm. In line with the pre-designed grating dimensions, a preset scanning control program of the 3D piezo-stage was worked out. Driven by the program, a series of grating units were precisely fabricated one after another along the axial direction of the microfiber, forming a periodic fully encircled μ-PFBG. Afterwards, the microfiber was immersed into isopropyl for 30 min, acetone for 1 min, and alcohol for 1 min, in sequence, and finally dried in air to remove any non-polymerized photoresist.

## 3. Results and Discussion 

The SEM image of this novel fully encircled μ-PFBG is shown in Figure 2a. The designed dimensions of the μ-PFBG are featured with a unit pitch of 1 µm, thickness of 300 nm, lateral dimensions of 3 μm × 3 μm, number of 60, and a diameter of the microfiber of approximately 2 μm. The number of units is related to the modulation depth of the grating. Too many units mean an increased risk of device consistency in nanofabrication. In order to maintain a better and consistent fabrication device, a proper number of units with effective resonance intensity are needed. The period of the polymer element affects the resonant wavelength position of the grating, while the thickness of the element affects the duty cycle in a cycle. The improvement of the thickness is of great significance for the further compression of the device size. Due to the size of the laser focus and the inherent energy jitter of the laser energy itself, the actual unit size is slightly fluctuated and slightly larger than the design size, which is in the range of 275 nm to 341.7nm. The transmission (red) and reflection (black) spectra of this μ-PFBG in the alcohol solution were both measured with a BBS and an OSA, as shown in Figure 2b. The resonance wavelength is approximately 1420 nm, with a reflection intensity of 15 dB. The resonance reflection wavelengths of two fully encircled μ-PFBG with different pitches of 1 µm and 1.5 µm are compared in Figure 2c. Both share identical fabrication parameters except for the unit pitch. Results show that the increasing unit pitch dimensions mainly affect the resonance position, shifting to a longer wavelength from approximately 1420 nm to 1550 nm, corresponding to intervals ranging from 1 µm to 1.5 µm, indicating that different resonance wavelength positions can be obtained by adjusting the pitch intervals. 

We investigated the ambient RI liquid sensitivity of the proposed fully encircled μ-PFBG at room temperature (25 °C). The novel μ-PFBG device was sequentially immersed in a series of the mixture of water and alcohol, whose RI values increased from 1.3325 to 1.343. The solution samples with different refractive indices were obtained by mixing alcohol with aqueous solution in different proportions, and adjusting the mixing ratio after calibration with a precise refractometer (Abbe 60, Bellingham, England). Figure 3a shows the relationship between Bragg resonance wavelength and different liquid RI mixtures. With the increasing ambient RI values, the resonance wavelength shifts linearly, as is seen in Figure 3b. The RI sensitivity is approximately 440 nm/RIU, which is much higher than the previously reported μ-PFBG, ranging from 30 nm/RIU to 229 nm/RIU [4,18,20,21,22]. 

Moreover, the characteristics of the strain and temperature responses were also studied, as illustrated in Figure 4a,b, respectively. In strain response measurement, the two fiber ends were fixed onto two micro-displacement platforms, and the resonance wavelength response to strain was measured by quantitatively changing the distance of the micro-displacement platform. As shown in Figure 4b, the Bragg resonance wavelength changes linearly to a longer wavelength from 1530 nm to 1540 nm, corresponding to the increasing strain from 0 µε to 1800 µε. In addition, the calculated strain sensitivity is approximately 6.3 pm/µε, which is higher than reported microfiber FBGs [11] in other works.

In temperature response experiments, the proposed fully encircled μ-PFBG was placed in a heating box, accompanied by a temperature controller. The temperature was continuously varied from 35 °C to 50 °C, and the resonance wavelength showed a linear relationship with the increasing temperature, as can be seen in Figure 4b. The temperature sensitivity of this novel μ-PFBG is approximately 88 pm/°C, which is higher than conventional FBGs [23]. The sensitivity of the strain and temperature advantages are mainly due to the size reduction of the grating period and unit size. The same tiny variations of strain and temperature show greater relative structural deformation on the ultra-small fully encircled μ-PFBG proposed in this work. On the other hand, polymer materials are more sensitive to stress and temperature than fiber silicon materials.

## 4. Conclusions

We demonstrated a fully encircled μ-PFBG device fabricated by femtosecond laser nanofabrication, featured with numbers of thin sheet units fully wrapped around the microfiber. The designed lateral dimension of the grating units (3 μm × 3 μm) was slightly larger than the diameter of the microfiber (~2 μm), to obtain an effective evanescent field modulation. The fully-wrapping-units design enhanced the stability of record thin grating units of 300 nm, by increasing the contact area with the microfiber surface, and was further consolidated by optimization of the photoresist composition. The proposed fully encircled μ-PFBG exhibits good transmission and reflection resonance characteristics, and the sensitivity to RI and strain and temperature were also studied. The RI, temperature, and strain sensitivity achieved were, respectively, 440 nm/RIU, 6.3 pm/µε and 88pm/°C, indicating some sensitivity improvements, compared with other similar works. This scheme exhibits more possibilities in complex integrated optical fiber device fabrication, and takes a further step toward more practical applications in various fields. 

## Figures and Tables

**Figure 1 materials-15-07753-f001:**
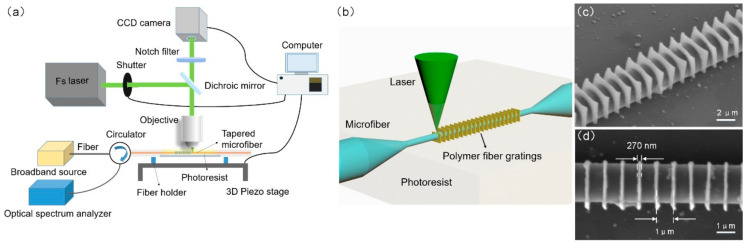
(**a**) Experimental setup of the femtosecond laser nanofabrication of the fully encircled μ-PFBG device. (**b**) Schematic diagram of the process of the novel μ-PFBG fabrication. (**c**,**d**) are the SEM images of this fully encircled μ-PFBG at angles of 45°and 0°, respectively.

**Figure 2 materials-15-07753-f002:**
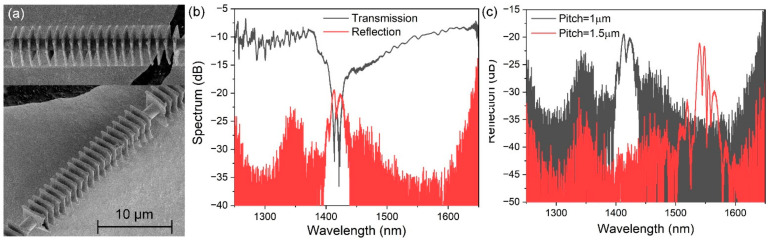
Characteristics of the proposed fully encircled μ-PFBG device. (**a**) The SEM images of the encircled μ-PFBG on a microfiber with a diameter of 2 µm. (**b**) The transmission and reflection spectrum in alcohol solution. (**c**) The comparison of the resonance reflection spectrums of the fully encircled μ-PFBG with the grating units pitch of 1 µm and 1.5 µm, with other dimensions being identical.

**Figure 3 materials-15-07753-f003:**
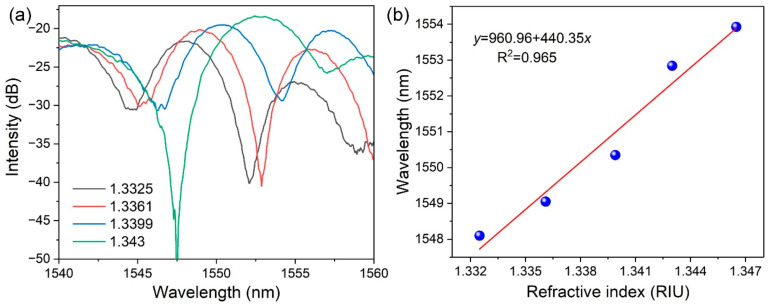
(**a**) Relationship between Bragg resonance wavelength and liquid RI. (**b**) The linear relationship of refractive index and resonance wavelength of the fully encircled μ-PFBG.

**Figure 4 materials-15-07753-f004:**
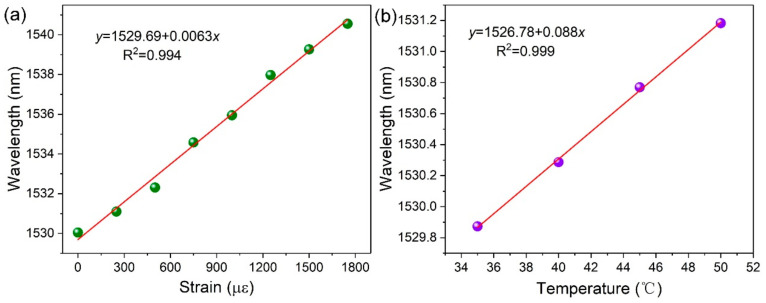
The Bragg resonance wavelength responses linearly to (**a**) strain and (**b**) temperature.

## Data Availability

The data presented in this study are available on request from the corresponding author.

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
