# Peer review of "A Fully-Encircled Polymerized Microfiber Bragg Grating by 3D Femtosecond Laser Nanofabrication"

_materials, 2022, doi:10.3390/ma15217753_

Round 1
Reviewer 1 Report
The authors present the fabrication by two photon polimerization of a microfiber Bragg grating. This result is achieved by selectively irradiating a monomer surrounding the microfiber, in such a way to create periodic structures leading to a modulation of the refractive index experienced by the evanescent field. The authors then test the sensitivity of this device with respect to the pitch of the microstructures, the refractive index of the environment, the applied strain, and temperature, finding good results in all the cases.
The paper is enjoyable to read, clear and well written. Moreover, it is scientifically sound, therefore I recommend its publication on Materials. I have only some minor comments:
1) Better words could be chosen in some parts instead of the used ones (e.g. "meanwhile" in line 12, "responsibility" in caption of Fig. 4), and some sentences should be revised (e.g. lines 32-33).
2) More details about the used fabrication parameters, such as pulse energy, scanning speed or used laser system, should be given in the Materials and Methods.
3) Is the flame-heating method for reducing the diameter of the fiber a home-made one, or it has been industrially developed? Maybe it should be specified in the main text.
4) The achieved resolution of 270-300 nm should be better compared with the state of the art for giving a better background.
Reviewer 2 Report
In the introduction, the authors could have included some advantages of using microfiber Bragg grating as compared to conventional fiber Bragg gratings. One example of information that may have been provided is whether the microfiber based Bragg gratings feature higher sensitivity than the conventional fiber Bragg gratings.
Is TPP the only fabrication method to obtain microfiber Bragg grating? If not, the other fabrication methods should be listed with references, and compared to TPP.
The authors mention the flame-heating method used to produce the microfibers, but do not provide further details about it. It would be useful to have a description of the method and to include references for it.
The authors explain that the polymer units featuring 270 nm of thickness were too thin and not strong enough, so an optimal thickness of 300 nm was tested and defined as a safer choice. Do the 270 nm-thick polymer units present significant differences in the optical properties as compared to the 300 nm ones? If not, it is unnecessary to include the results for it.
Fig. 3 could have been included as an item c in Fig. 2.
A microfiber Bragg grating with 60 polymer units with thickness of 300 nm and pitch of 1 μm is reported as a proper choice in the article, but the authors did not comment on the impact of the number of polymer units and their thickness on the device’s performance. For example, which impact will it have to increase the polymer unit thickness?
What was the criterion to determine the spectral position of the Bragg resonance wavelength? Some of the resonances seem a bit broad, so I am not sure whether a find-peaks algorithm will be suited for this.
The authors mention that the novel μ-PFBG device was sequentially immersed in a series of mixtures of water and alcohol, whose RI values increased from 1.3325 to 1.343. How were these refractive indices determined? It is important to include this information in the manuscript.
The authors claim that the strain sensitivity and temperature sensitivity of their μ-PFBG is higher than that of previously reported μ-PFBG and conventional FBGs, respectively. Why is that? What makes their device perform better?
